# Novel Aptamer Strategies in Combating Bacterial Infections: From Diagnostics to Therapeutics

**DOI:** 10.3390/pharmaceutics16091140

**Published:** 2024-08-29

**Authors:** Zijian Ye, Huaizhi Chen, Harrie Weinans, Bart van der Wal, Jaqueline Lourdes Rios

**Affiliations:** 1Department of Orthopedics, University Medical Center Utrecht, 3584 CX Utrecht, The Netherlands; z.ye@umcutrecht.nl (Z.Y.); c.huaizhi@umcutrecht.nl (H.C.); h.h.weinans@umcutrecht.nl (H.W.); b.c.h.vanderwal@umcutrecht.nl (B.v.d.W.); 2Department of Biomechanical Engineering, Faculty of Mechanical Engineering, Delft University of Technology (TU Delft), 2628 CD Delft, The Netherlands

**Keywords:** aptamers, bacterial infections, antimicrobial resistance, biosensors, biofilm

## Abstract

Bacterial infections and antimicrobial resistance are posing substantial difficulties to the worldwide healthcare system. The constraints of conventional diagnostic and therapeutic approaches in dealing with continuously changing infections highlight the necessity for innovative solutions. Aptamers, which are synthetic oligonucleotide ligands with a high degree of specificity and affinity, have demonstrated significant promise in the field of bacterial infection management. This review examines the use of aptamers in the diagnosis and therapy of bacterial infections. The scope of this study includes the utilization of aptasensors and imaging technologies, with a particular focus on their ability to detect conditions at an early stage. Aptamers have shown exceptional effectiveness in suppressing bacterial proliferation and halting the development of biofilms in therapeutic settings. In addition, they possess the capacity to regulate immune responses and serve as carriers in nanomaterial-based techniques, including radiation and photodynamic therapy. We also explore potential solutions to the challenges faced by aptamers, such as nuclease degradation and in vivo instability, to broaden the range of applications for aptamers to combat bacterial infections.

## 1. Introduction

Bacterial infections pose a substantial risk to worldwide public health, causing almost 7.7 million deaths each year. This accounts for 13.6% of all global deaths and 56.2% of fatalities attributable to sepsis [1]. The main cause of most of these deaths is linked to five bacterial pathogens: *Staphylococcus aureus* (*S. aureus*), *Escherichia coli* (*E. coli*), *Streptococcus pneumoniae* (*S. pneumoniae*), *Klebsiella pneumoniae* (*K. pneumoniae*), and *Pseudomonas aeruginosa* (*P. aeruginosa*). These bacteria are known for their ability to form biofilms, complex communities that adhere to surfaces and contribute to persistent infections [2,3]. Biofilms are responsible for over 80% of microbial infections, over 60% of hospital-acquired infections, and 60% of all chronic infections, according to the National Institutes of Health and the Centers for Disease Control and Prevention (project announcements PA-03-047 and PA-06-537) [4]. Biofilms also contribute to the escalating prevalence of antimicrobial resistance (AMR), which is projected to surpass cancer as the primary cause of death by 2050, with an estimated 10 million deaths per year [5]. This emphasizes the pressing need for effective diagnostic and therapeutic strategies to address bacterial infections and reduce their significant impact on global health.

The emergence of aptamers—short, single-stranded DNA or RNA molecules selected for their ability to bind specific targets with high affinity and specificity—has opened new avenues in the field of biomedical research. First introduced by Ellington and Gold in the 1990s, aptamers are screened through an in vitro selection process known as Systematic Evolution of Ligands by Exponential Enrichment (SELEX) [6,7]. Aptamers function similarly to antibodies as molecular recognition elements but offer several advantages [8]. They have the ability to adopt distinct three-dimensional conformations, enabling them to selectively and strongly interact with certain targets. Their binding capabilities make them suitable for a wide range of applications, such as diagnostics, therapy, imaging, and drug delivery systems [9,10]. Moreover, aptamers are also being evaluated as potential antimicrobial agents [11]. Although aptamers show great promise, their use in fighting bacterial infections is still in the early phases of development. This review specifically examines the application of aptamers in the diagnosis and therapy of bacterial infections.

## 2. Aptamers

The term “aptamer” is a compound of two Latin words: “aptus”, which means “fit” or “adapt”, and “meros”, which means “part” or “unit” [7]. This term was coined to describe the ability of these molecules to adapt and bind tightly to specific molecular targets. Aptamers are typically 10 to 100 nucleotides (nt) in length, and due to the complementary nature of their base pairs, they usually have unique three-dimensional structures. They are a combination of loops, stems, hairpins, pseudoknots, bulges, and G-quadruplexes, which are essential for target molecule recognition [12,13,14,15,16]. The aptamer–target contact is promoted by hydrogen bonds, electrostatic forces, and van der Waals interactions, leading to the formation of a stable complex [8]. Early research on aptamers mainly focused on RNA, as it was thought to create more complex and diverse three-dimensional structures than DNA [17,18]. However, later research shifted its attention to DNA aptamers, which were discovered to have similar selective binding capabilities despite previous beliefs. DNA aptamers also possess better stability and longer plasma half-lives when compared to RNA aptamers, making them well suited for in vivo applications [10,18]. 

Aptamers have numerous advantages compared to antibodies, including the ability to achieve comparable or even heightened specificity under optimal selection conditions, cost-effectiveness, extended shelf life, effortless modifiability, reduced immunogenicity, and improved tissue penetration owing to their smaller dimensions [19]. While antibodies are known for their high specificity, especially for complex protein targets, aptamers can be designed through the SELEX process to match or exceed this specificity, particularly for small molecules or closely related targets such as protein isoforms [20]. In addition, aptamers possess greater stability, especially with modifications, and demonstrate a high level of uniformity between batches because they are chemically synthesized and enriched by in vitro SELEX methods. Conversely, antibodies are proteins produced and cloned from recombinant cell lines, which have higher molecular weights, are more expensive to produce, and are prone to denaturation and degradation. Due to the complexity of the bioproduction process, including variations in cell growth conditions, post-translational modifications, and purification processes, the consistency of antibodies may vary from batch to batch. And the initial selection of antibodies typically involves the use of cellular immune responses in animal hosts. In addition, aptamers have the ability to target a wider variety of molecules, such as tiny molecules and ions, whereas antibodies are restricted to immunogenic compounds [19,21]. Table 1 presents a comparison of the advantages and disadvantages of aptamers and antibodies.

The gold standard method for aptamer selection is known as SELEX (Systematic Evolution of Ligands by Exponential Enrichment) [22,23]. The first step involves creating an aptamer library with 10^13^ to 10^15^ random nucleotide sequences produced via chemical synthesization, fragments of genomic DNA, or combinations. A classic SELEX library has two fixed ends for reverse transcription PCR (RT-PCR) and a random functional region ranging from 20 to 60 nucleotides [6]. The library is incubated with the target molecule, allowing binding. Unbound sequences are washed away, leaving only target-bound aptamers. These are released by thermal or chemical methods and amplified by PCR to create a new library for the next round. This process is repeated for 8 to 15 rounds, increasing the proportion of high-affinity aptamers and the specificity of the aptamer library. Washing stringency is increased each round to enhance specificity. Negative selection may be used to reduce nonspecific binding. Finally, aptamer sequences are cloned, sequenced, and evaluated for binding affinity, specificity, and stability [22,23]. Figure 1 depicts the standard SELEX process, illustrating its repetitive cycles of selection and amplification.

Over the years, the SELEX process has evolved to enhance aptamer selection, efficiency, specificity, and applicability. Figure 2 provides a concise overview of the common variations of SELEX. Various SELEX variations, including Cell-SELEX [24], In Vivo SELEX [25], Toggle-SELEX [26], and automated SELEX [27], target different aspects such as whole cells, living organisms, and closely related targets to improve selection efficiency. The initial library can include chemically modified nucleotides, multiple pools, or genomic DNA. Binding to targets is processed by various methods such as immobilization on magnetic beads [28], capillary electrophoresis [29], electrophoretic mobility shift assays (EMSAs) [30], fluorescent-activated cell sorting (FACS) [31], and surface plasmon resonance (SPR) [32]. The selection of the appropriate SELEX is essential for the identification of an aptamer that possesses the desired binding characteristics. Nevertheless, success is contingent upon the initial library, target, SELEX condition (such as medium), aptamer amplification procedure, and sequence analysis [33].

## 3. Aptamers as Diagnostic Tools for Bacterial Infections

In clinical settings, the diagnosis of bacterial infections continues to be a significant challenge. The drawbacks of traditional methods include their time-consuming nature, their low sensitivity, and the necessity of sophisticated apparatuses and trained personnel, such as in culture-based techniques, polymerase chain reaction (PCR), and immunoassays [34,35]. These drawbacks can result in delayed diagnosis and treatment, potentially worsening patient outcomes and leading to the spread of infections. Aptamers have emerged as promising molecular recognition elements that can enable early diagnosis by integrating with various biosensors and imaging technologies [36,37,38]. 

### 3.1. Aptamer-Based Biosensors

Aptamer-based biosensors (aptasensors) can enable the detection of bacterial infections by combining specific nucleic acid aptamers and signal conversion elements [39]. The aptasensor architecture typically consists of a recognition element (aptamer), a conversion element, and a signal processing unit [40]. Upon binding of the aptamer to the target, a physical or chemical change occurs in the conversion element, generating a detectable signal. Aptasensors can utilize electrochemical, optical, or mass transduction methods, employing either label-free or label-based methodologies [41] (Figure 3). This design enables aptasensors to provide highly sensitive and specific detection [42]. Despite their extensive use in food safety and environmental monitoring, aptasensors are relatively novel in the domain of bacterial infection detection. The aptasensors that have been recently designed for the clinical detection of bacterial cells or their toxins are summarized in Table 2.

#### 3.1.1. Mass-Sensitive Aptasensors

Among the several types of mass-sensitive aptasensors, quartz crystal microbalance (QCM) aptasensors are noteworthy. Their working principle is based on the change in resonance frequency [53]. Based on this principle, Ji et al. have developed a label-free, highly sensitive SH-SAW aptasensor for endotoxin detection using a single layer of graphene (SLG) grown via chemical vapor deposition (CVD) [43]. The endotoxin-specific aptamers were covalently attached to the surface, with a detection limit of 3.53 ng/mL, effectively discriminating endotoxins from *P. aeruginosa*-derived aflatoxins. This approach is potentially applicable for endotoxin detection in the clinic. Another report detailed the development of a graphene-based aptasensor combined with interdigitated gold electrodes (IDEs) on a series of piezoelectric quartz crystals (SPQCs) [44]. This aptasensor has the ability to quickly and accurately identify *S. aureus* within a time frame of 60 min, making it well suited for clinical diagnostics.

#### 3.1.2. Electrochemical Aptasensors

Electrochemical aptasensors function by detecting electrical signals produced by metabolic processes. These sensors are appealing because of their exceptional sensitivity, ease of use, and ability to be made smaller for use in portable devices [54]. *Mycobacterium tuberculosis* (*M. tb*) is a major cause of tuberculosis, requiring rapid and accurate detection. Das et al. demonstrated a novel aptamer-based electrochemical biosensing strategy for detecting *M. tb* [45]. This method employs the target-induced structural conversion of the H63 SL-2 M6 aptamer and the electroactivity of aptamer-labeled methylene blue (MB) to detect HspX in cerebrospinal fluid samples. This strategy achieved a sensitivity of approximately 95% and a specificity of around 97.5% within 30 min. Another electrochemical aptasensor was created to detect MPT64, a protein secreted exclusively by *M. tb*, in human serum at an ultrasensitive level [46]. This aptasensor employs a capture aptamer (MBA I) that is immobilized on a gold electrode and a signal aptamer (MBA II) that is labeled with gold nanoparticles on a fullerene-doped polyaniline (C60-PAn) redox nanoprobe (Figure 4). In serum samples from patients with tuberculosis, this aptasensor exhibited superb specificity and sensitivity, as well as a detection limit of 20 fg/mL.

*P. aeruginosa* is a prevalent bacterium linked to nosocomial illnesses [55]. Shahrokhian et al. created a very sensitive diagnostic tool to detect the presence of *P. aeruginosa* at the cellular level [47]. This device utilizes aptamers that are immobilized on the surface of engineered zeolitic imidazolate framework-8 (ZIFs-8) (Figure 5A). This platform employs ferrocene–graphene oxide (Fc-GO) as an electroactive indicator, which is released from the electrode surface upon binding with *P. aeruginosa* (Figure 5B), allowing detection with a limit as low as 1 CFU/mL, and has shown excellent recovery rates from spiked human urine samples.

*S*. *aureus* is a significant cause of various infections, including antibiotic-resistant strains like methicillin-resistant *Staphylococcus aureus* (MRSA). A selective sandwich detection method combining dual aptamers and antibiotics has been developed for enhanced specificity and sensitivity [48]. The technique employs vancomycin, which adheres to the cell wall of gram-positive bacteria, immobilized on a carbon electrode. This arrangement effectively concentrates bacterial cells in under 10 min. This aptasensor can identify *S. aureus* and *Bacillus cereus* in milk or bovine serum within 45 min, with a detection limit of 100 CFU/mL. Zelada-Guillén et al. reported a real-time, label-free method for detecting *S. aureus* on skin [49] (Figure 6). They utilized pyrene/amine-modified aptamers specific to *S. aureus* and single-walled carbon nanotubes (SWCNT) as potentiometric aptasensors, achieving a sensitivity of 1.52 mV/Decade but an upper detection limit of 10⁷ CFU/mL and a lower detection limit of 8 × 10² CFU/mL and a sensitivity of 0.36 mV/Decade, respectively. 

#### 3.1.3. Optical Aptasensors

Optical aptasensors provide several benefits, including their exceptional sensitivity, convenient accessibility, compact size, affordability, and ability to detect substances in real time without the need for labeling. They function by using several optical principles, such as colorimetric, fluorescent, surface-enhanced Raman scattering (SERS), and surface plasmon resonance (SPR) principles [56,57].

A *S. aureus* dual-recognition ratiometric fluorescence nano aptasensor has been developed [50]. This aptasensor identifies cell-level infections because of its large Stokes shift. The system uses blue fluorescent carbon nanoparticles (CNPs) with copious π-electrons as energy sources. Near-infrared (NIR) fluorescent quantum dots (QDs) tagged with aptamers and vancomycin are used for titration. It accurately detects *S. aureus* with great specificity and can identify *S. aureus* cells at 1.0 CFU/mL.

Zhang et al. developed a dual-recognition platform using vancomycin and aptamers to detect both *E. coli* and *S. aureus* simultaneously [51] (Figure 6). Vancomycin, when integrated into Fe_3_O_4_@Au nanoparticles, functions as a versatile bacterial capture agent, efficiently concentrating the desired germs. Aptamer-functionalized gold nanoparticles are utilized, along with two distinct forms of SERS tags, to perform precise and sensitive quantitative analysis of the target microorganisms. This platform exhibits a detection threshold of 20 cells/mL for *S. aureus* and 50 cells/mL for *E. coli*.

A microfluidic device with integrated functionality has been developed for the detection of *Acinetobacter baumannii*, highly dangerous bacteria that are resistant to several drugs and are mostly responsible for hospital-acquired infections [52]. This detection approach utilizes two aptamers: one connected to magnetic beads to collect bacterial cells and another labeled with quantum dots for quantification. The microfluidic system, which is powered by electromagnetism and includes a module that uses a light-emitting diode to create fluorescence, is capable of detecting as few as 100 CFU per reaction in under 30 min. This process can be completely automated without any human interaction or the requirement of external equipment.

### 3.2. Aptamer Molecular Imaging

Aptamers have the potential to revolutionize in vivo imaging methodologies, specifically in the areas of real-time disease detection and therapeutic monitoring, such as in cancer and bacterial infections [58]. Linkers can be used to conjugate these molecules with fluorescent markers or radioactive isotopes, which can then be administered intravenously [59]. A portion of these conjugated aptamers localizes to the target site after being administered, while the remaining portion is excreted from the body. Various imaging systems, including fluorescence imaging, MRI (magnetic resonance imaging), SPECT (single-photon emission computed tomography), and PET (positron emission tomography) scans, can be employed to observe this selective localization and clearance [60]. Figure 7 illustrates various molecular imaging systems based on aptamers.

#### 3.2.1. Fluorescence Imaging

Fluorescence imaging with aptamers enables imaging of biological processes at the cellular and molecular levels. Du et al. developed a nanoprobe called AuNS-Apt-Cy that is responsive to gelatinase [61]. This nanoprobe is used for in situ NIR fluorescence imaging and localized photothermal therapy (PTT) of MRSA infections. This nanoprobe is functionalized with an MRSA-specific aptamer and a gelatinase-responsive heptapeptide linker (CPLGVRG) conjugated with cypionate dye on gold nanostars. In aqueous environments, the nanoprobe is non-fluorescent due to fluorescence resonance energy transfer (FRET) between the gold nanostar core and the cypionate dye. When encountering MRSA, the CPLGVRG linker is cleaved, activating NIR fluorescence and allowing the nanoprobe to accumulate at the infection site. This enables both fluorescence imaging and PTT by disrupting bacterial cell walls and membranes. In vivo experiments demonstrated rapid and sensitive NIR fluorescence imaging in diabetic wound and implanted bone plate mouse models, detecting as few as 10⁵ colony-forming units [61]. However, fluorescence imaging is restricted by inadequate tissue penetration as a result of photon absorption and scattering [62].

#### 3.2.2. Nuclear Medicine Imaging

Nuclear medicine imaging is capable of highly sensitive detection of infection sites and deep tissue visualization [63,64]. The application of radiolabeling techniques to aptamers for nuclear medicine imaging is a significant advancement in the imaging field. This technique enables aptamers to bind to radioisotopes used in PET or SPECT imaging, such as technetium-99m, iodine-131, fluorine-18, copper-64, gallium-68, and zirconium-89 [36,65].

Technetium-99m (^99m^Tc) is widely utilized in around 80% of nuclear medicine procedures worldwide [66]. It is preferred due to its emission of gamma rays with an energy of 140 keV, its convenient half-life of 6 h, and its availability from Mo-99/Tc-99m generators. Lacerda et al. labeled aptamers seq6 and seq30, targeting (1→3)-beta-D-glucan in the bacterial cell wall, with ^99m^Tc to visualize *S. aureus* infection foci in mice using SPECT [67]. In infected animals, the uptake of the ^99m^Tc-labeled aptamers in the infected thigh was significantly higher than in the non-infected muscle. The target/non-target ratios for seq6 and seq30 were 3.17 ± 0.22 and 2.66 ± 0.10, respectively, outperforming the radiolabeled library control. This method effectively imaged infection sites specifically infected with *S. aureus*. Additionally, the same research group reported that Antibac1, an aptamer targeting peptidoglycan, was successfully labeled with ^99m^Tc to distinguish between bacterial and fungal infections in mice [68].

#### 3.2.3. Magnetic Resonance Imaging (MRI)

MRI employs powerful magnetic fields and radio waves to provide precise visual representations of internal organs and tissues. Aptamers have been linked with paramagnetic substances such as gadolinium-containing compounds and superparamagnetic iron oxide nanoparticles (SPIONs) for the purpose of molecular imaging by MRI [18]. Wang et al. merged an A10 aptamer, which specifically targets PSMA on prostate cancer cells, with SPIONs [69]. The A10-SPION combination exhibited exceptional sensitivity and selectivity in its ability to target prostate cancer cells that express PSMA, thus establishing its efficacy as an MRI imaging agent. This principle exhibits the potential for visualizing bacterial infections as well.

## 4. Aptamer-Based Therapies for Bacterial Infections

Aptamer-based therapies commonly utilize one of the following two approaches: (1) Aptamers can act as modulators by either inhibiting interactions between disease-related targets, such as protein–protein or receptor–ligand interactions, or by activating the function of target receptors. (2) Cell-type-specific aptamers can serve as carriers, delivering other therapeutic agents to specific cells or tissues. Below is a brief overview of how aptamers have been used in the infection field. 

### 4.1. Therapeutic Aptamers

#### 4.1.1. Inhibiting Bacterial Growth and Reproduction

Aptamers, as novel antibacterial agents, can function through various mechanisms to inhibit bacterial growth or disrupt key bacterial functions [70,71,72]. Aptamers can disrupt essential cellular processes by binding to key bacterial proteins, signaling molecules, or other critical components. For example, aptamers have been developed to target the BamA protein in *P. aeruginosa*, a crucial component of the outer membrane protein assembly complex [73]. By binding to BamA, the aptamer blocks the insertion and assembly of outer membrane proteins, disrupting the integrity of bacterial membranes and leading to cell death due to leakage of cellular contents. Additionally, the selected aptamers can inhibit bacterial growth by targeting key enzymes such as inorganic polyphosphates (PolyPs) involved in bacterial persistence, stress response, and virulence [72]. These RNA aptamers can inhibit polyphosphate kinase 2 of *M*. *tb*, disrupting polyphosphate-dependent processes in pathogenic organisms.

#### 4.1.2. Blocking Virulence Factors

Bacteria-produced toxins can either directly destroy host cellular structures or indirectly cause disease-related effects by activating the host’s immune response [74]. Aptamers have demonstrated promise as highly efficient molecules for specifically binding and neutralizing bacterial toxins. For example, RNA aptamers have been created to hinder the attachment of anthrax toxin, produced by *Bacillus anthracis*, to its receptor [75]. These aptamers exhibit a strong affinity for binding and possess the capability to counteract the toxin’s effects in macrophages. Vivekananda et al. isolated several DNA aptamers that specifically inhibited the cytotoxic activity of α-toxin produced by *S*. *aureus*, and four of them significantly inhibited α-toxin-mediated cell death in Jurkat T cells [76]. Two of the aptamers also inhibited α-toxin-induced upregulation of the inflammatory cytokines TNF-α and IL-17.

#### 4.1.3. Inhibiting Bacterial Attacks on Immune Cells

Macrophages and monocytes are critical components of the host immune defense against bacterial infections [77]. Intracellular pathogens, such as *M. tb* and *Salmonella typhimurium* (*S. typhimurium*), have evolved strategies to invade and replicate within these immune cells, evading host defenses [78]. Aptamers targeting specific proteins on the surface of these bacteria can inhibit their invasion of immune cells. For example, aptamers targeting *M. tb* H37Rv specifically inhibit its invasion of macrophages by binding to the outer membrane proteins and stimulating macrophages to secrete pro-inflammatory cytokines like IFN-γ, IL-15, and IL-17, enhancing immune defense [79]. Similar findings suggested that an aptamer called ZXL1 specifically binds to a component of *M*. *tb* called mannose-capped arabinogalactan (ManLAM), which inhibits the entry of *M. tb* into macrophages [80]. Pan et al. identified an RNA aptamer (S-PS 8.4) that binds to *S. typhimurium* type IVB pili, significantly inhibiting the entry of pili strains into human THP-1 cells [81].

#### 4.1.4. Inhibiting Biofilm Formation

Biofilms are organized communities of microorganisms, predominantly bacteria that are affixed to surfaces and enclosed in a self-generated extracellular matrix (ECM) [82]. These biofilms are highly resistant to antibiotics and the immune system, which presents a significant challenge in the treatment of chronic and hospital-acquired infections. Additionally, they provide structural integrity and protection to microbial communities [83]. Aptamers have shown great potential in disrupting biofilm formation by targeting key components involved in biofilm formation and maintenance.

Aptamers can effectively inhibit bacterial adhesion, a critical initial step in biofilm formation. A pioneering study developed a bifunctional aptamer/berberine-loaded graphene oxide composite to target MRSA biofilms [84]. Aptamer 1, selected for its high affinity for penicillin-binding protein 2a (PBP2a), a key factor in MRSA biofilm formation, effectively inhibits biofilm formation by reducing cell adhesion and downregulating the expression of the accessory gene regulator (agr) involved in quorum sensing. Furthermore, R8-su12 RNA aptamers, which are bound by polysaccharide intercellular adhesin (PIA) located on the cell surface, have been shown to efficiently inhibit the formation of biofilms in *S. suis* serotype 2 and strain P1/7 by up to 61.2% in vitro [85].

Aptamers can effectively suppress biofilm development by interfering with bacterial quorum sensing. Quorum sensing is a mechanism used by bacteria to communicate with each other and coordinate the activation of genes involved in the formation of biofilms [86]. By targeting quorum-sensing molecules or their receptors, aptamers can disrupt the communication necessary for biofilm formation and maintenance. The aptamer PmA2G02 has shown efficacy in inhibiting biofilm formation, swarming motility, and cell viability in *Proteus mirabilis* by specifically targeting proteins that are crucial for adhesion, motility, and quorum sensing [87]. This in vitro study demonstrated that the application of aptamer therapy led to a 50% decrease in the thickness of the biofilm and increased dispersion compared to the untreated control group. Similarly, the specific aptamer SELEX 10 Colony 5, identified by cell SELEX, effectively inhibits biofilm formation of enteropathogenic *Escherichia coli* (EPEC) strain K1.1 in vitro by interfering with movement and reducing mRNA levels of related genes [88].

Aptamers can target and disrupt the structural components of biofilms by binding to key molecules within the biofilm matrix. This approach compromises the integrity of the biofilm, making the embedded bacteria more vulnerable to treatment and enhancing the effectiveness of antibiotics against biofilm-associated infections. Ning et al. reported that aptamer 3, which targets *Salmonella choleraesuis*, causes most bacteria to remain in a planktonic state, thereby increasing the efficacy of ampicillin sodium [89]. Shatila et al. demonstrated that the three-dimensional structure of the biofilm is altered by the DNA aptamer Apt17, which targets *Salmonella* protein A (SipA) [90]. The treated cells lost the ECM and, when combined with ampicillin, significantly reduced biofilm formation in vitro.

### 4.2. Aptamer–Nanomaterial Conjugates

Recent developments in the application of aptamer–nanomaterial conjugates have demonstrated significant potential in the prevention of bacterial infections. The precise targeting ability of aptamers and the unique properties of nanomaterials are combined in these multifunctional molecules, improving the delivery and efficacy of antimicrobial agents [91,92]. The potential and applications of these conjugates are highlighted in this section, which explores a variety of innovative strategies and recent studies based on their antibacterial mechanisms and effects.

#### 4.2.1. Reactive Oxygen Species (ROS) Generation

Aptamer–nanomaterial complexes have demonstrated potent bactericidal effects through the generation of reactive oxygen species (ROS). ROS are molecules with a high reactivity that can cause harm to different parts of cells, such as DNA, proteins, and lipids, resulting in cell death [93]. Single-atom nano-enzymes (SAzymes) based on platinum-modified carbon nitride nanorods have demonstrated enhanced peroxidase-like activity, generating hydroxyl radicals that effectively inactivate gram-negative bacteria [94]. The modulation by aptamers ensures high sensitivity to antibiotics, illustrating the potential of SAzymes in antibiotic detection, susceptibility testing, and wound healing in bacterial infections in mouse models.

#### 4.2.2. Nanoparticle-Loaded Conjugates

Nanoparticles, ranging from 1 to 100 nanometers in size, possess unique properties, making them effective for biomedical applications like drug delivery [95]. These characteristics encompass a substantial ratio of surface area to volume and the capability to permeate biological membranes. In antibacterial therapy, nanoparticles functionalized with antimicrobial agents enhance targeting and efficacy. Ucak et al. reported that aptamer-functionalized nanoparticles loaded with ticlopidine significantly reduce the minimum inhibitory concentration of the antibiotic against multi-drug-resistant *S. aureus*, allowing for more efficient inhibition at reduced doses [96].

Antimicrobial peptides (AMPs) are small proteins that can kill bacteria by disrupting their cell membranes, interfering with cellular processes, or modulating the host immune response [97,98]. The high specificity of aptamers allows for the targeted delivery of AMPs, enhancing their effectiveness against bacterial infections. The high specificity of aptamers selected from the fGmH RNA library for *Staphylococcus aureus* protein A (SpA) allows for targeted delivery of AMP-functionalized silver nanoparticles (AgNPs) to effectively combat *S. aureus* infections [99]. Another study demonstrated that gold nanoparticle–DNA aptamer conjugates can effectively deliver antimicrobial peptides (AMPs) to combat *Vibrio vulnificus* infections [100]. This approach significantly reduced bacterial colonization in vivo and ensured a 100% survival rate of the infected mice.

Recent research has also demonstrated that DNA aptamer-functionalized gold nanoparticles (AuNP-Apt) can effectively deliver AMPs to mammalian biological systems, enhancing the stability of AMPs [101]. The study involved the attachment of C-terminal hexahistidine-tagged A3-APO (A3-APO His) AMPs to AuNPs (AuNP-Apt His) that were linked to DNA aptamers carrying His tags. The *S. typhimurium*-infected HeLa cells were administered with these nanoparticles that were loaded with substances. The AuNP-Apt His, which contains A3-APO His, effectively prevented organ colonization in mice infected with *S. Typhimurium*, leading to a 100% survival rate.

#### 4.2.3. Biofilm Disruption

As mentioned above, biofilms are a challenge in treating bacterial infections. Aptamer–nanomaterial conjugates offer promising solutions to this problem. Silver nanoparticle–aptamer complexes have exhibited significant anti-biofilm activity against *S*. *aureus* and *Streptococcus pyogenes* on various substrates, including titanium, enhancing the ability to disrupt biofilms and reduce bacterial viability within these structures [102]. DNA-templated silver nanoclusters (DNA-AgNCs) combined with bacterial aptamers enable visual monitoring and effective elimination of bacteria. For example, DNA-AgNCs linked to *S. aureus* aptamers achieved fluorescence enhancement and promoted antibacterial effects [103]. Furthermore, aptamer-conjugated SWCNT and aptamer–ciprofloxacin–SWCNT conjugates have shown remarkable efficacy in suppressing the biofilm formation of *P. aeruginosa*, exploiting the physical and chemical properties of SWCNTs to disrupt bacterial cell walls and inhibit biofilm development [104]. Aptamer-targeted liposomes and mesoporous silica nanoparticles have also been successful in eradicating *S. aureus* biofilms, highlighting the potential of these approaches to treat persistent biofilm-associated infections [105,106].

#### 4.2.4. Advanced Platforms

Advanced platforms have been developed to further enhance the efficacy and targeting capabilities of aptamer–nanomaterial conjugates. The GATC therapeutic platform, consisting of a G-quadruplex/heme DNAzyme aptamer probe (single-stranded DNA molecules with catalytic function), tannic acid-chelated gold nanoparticles, and copper-based metal–organic framework (MOF) nanosheets, has demonstrated significant bactericidal activity against MRSA at low doses [107]. DNA origami nanostructures functionalized with aptamers have been successfully developed to deliver lysozyme, an antibacterial enzyme, in a targeted and effective manner [108] (Figure 8). These structures have shown greater efficacy in inhibiting bacterial growth compared to free lysozyme treatment, tested against both gram-positive (*Bacillus subtilis*) and gram-negative (*E*. *coli*) bacteria.

The exploration of aptamer-gated nanocapsules has unfolded novel avenues for antibiotic delivery systems, specifically against *S*. *aureus* infections [109]. Conjugates loaded with vancomycin have exhibited specific targeting capabilities while maintaining low toxicity to other bacterial strains, such as *Staphylococcus epidermidis*. These findings are complemented by the use of conjugation of vancomycin-loaded nanoparticles with the cyclic 9-amino-acid peptide CARGGLKSC (CARG), which have shown tenfold greater effectiveness against staphylococcal lung infections compared to free vancomycin in mouse models [110].

#### 4.2.5. Photodynamic Therapy (PDT)

Aptamer-based photodynamic therapy (PDT) is a potential approach for treating bacterial infections. This method harnesses the strong selectivity of aptamers to transport therapeutic chemicals that generate ROS and specifically eliminate pathogens [111].

Nanomaterial conjugates are essential for augmenting the efficacy of PDT. Song et al. synthesized titanium dioxide (TiO_2_) particles that were linked to a mixture of single-stranded DNA aptamers specific to *Escherichia coli*. This was performed to improve the disinfection process [112]. The TiO_2_-Apc required a significantly smaller amount of material to achieve approximately 99.9% inactivation of *E. coli* at a concentration of 10^6^ CFU/mL. Furthermore, graphene oxide nanoparticles (NGO) that have been modified with DNA aptamers have been utilized in a targeted biodiagnostic system for antimicrobial photodynamic therapy against *Porphyromonas gingivalis* [113]. This technology effectively binds pathogens, produces ROS when activated by light, and greatly decreases the viability and production of biofilms by *Porphyromonas gingivalis*.

An alternative and creative method involves utilizing DNA aptamer-conjugated magnetic graphene oxide (Apt@MGO) as a biocompatible nanoplatform to specifically and quickly eliminate MRSA [114]. Apt@MGO specifically attacks MRSA cells and, when exposed to a NIR laser, transforms light into heat, resulting in localized heating and the death of the cells. The study showed that Apt@MGO effectively deactivated over 78% of scattered MRSA cells and more than 97% of clustered MRSA cells within 200 s of NIR exposure. Moreover, the antimicrobial properties of antimicrobial PDT can be intensified by incorporating the dermcidin-derived peptide DCD-1L onto aptamer-functionalized emodin nanoparticles (Apt@EmoNp-DCD-1L) [115]. When used together with a blue laser, this method enhances the effectiveness of PDT in fighting against the biofilm of *Enterococcus faecalis*, and it dramatically reduces the expression of genes that are responsible for the production of bacterial biofilms.

Furthermore, Wang et al. created a sophisticated nanoplatform called ICG@GO-Apt, which combines aptamers (Apt), indocyanine green (ICG), and carboxyl-functionalized graphene oxide (GO-COOH) [116]. This nanoplatform was developed with the specific purpose of eradicating *S. typhimurium* biofilms. Aptamer-functionalized nanosheets (NSs) specifically cluster in the vicinity of abscesses generated by pathogens, hence increasing drug concentration and enabling precise drug administration. ICG@GO-Apt generates heat and reactive oxygen species when exposed to near-infrared light. The phototherapy platform demonstrated excellent biocompatibility and achieved a remarkable biofilm removal rate of over 99.99% in an abscess formation mouse model.

### 4.3. Aptamer-Mediated Immunomodulation

Aptamers can be utilized for immunotherapeutic treatment of bacterial infections by regulating the immune response. Researchers have studied the potential of enhancing the immune system to fight infections and reduce harmful inflammatory responses. This is especially crucial in illnesses like sepsis when an excessively strong immune response can be just as harmful as the damage caused by bacteria [117]. Figure 9 illustrates the principle of aptamer-mediated immunomodulation.

#### 4.3.1. Regulation of Innate Immune Responses

Toll-like receptors (TLRs) are crucial components of the innate immune system, responsible for detecting certain chemical patterns associated with infections and triggering immunological responses [118]. Aptamers targeting TLRs have shown great potential in regulating these pathways. The TLR2 antagonist aptamer AP177 has been shown to significantly inhibit NF-kB activity induced by specific TLR agonists, thereby reducing inflammation induced by gram-positive bacterial infections [119]. Moreover, DNA aptamers that target the lipopolysaccharide (LPS) of *Escherichia coli* O111, when coupled with human C1qrs protein, induce significant bacterial lysis in the presence of human serum through a competitive alternative pathway that avoids complement activation [120].

#### 4.3.2. Enhancement of Adaptive Immune Responses

Aptamers are great for boosting adaptive immune responses. The aptamer NK2 has strong binding and selectivity towards virulent strains of *M. tb* (H37Rv) [121]. These aptamers stimulate the production of defensive cytokines, such as IFN-γ, by CD4+ T cells. Consequently, the survival rate of *M. tb*-infected mice increased, and the number of bacteria in their spleens decreased when compared to control mice.

#### 4.3.3. Targeting Immune Checkpoints

Immune checkpoints are regulatory molecules located on immune cells that, when activated, can improve immune functions [122]. While these immune inhibitors can reduce immune pathology or autoimmunity, they also impede immune-mediated clearance of infections, which contributes to the persistence of the disease. Monoclonal antibodies and aptamers, which are examples of checkpoint inhibitors, prevent these checkpoint molecules from interacting with their ligands, thereby restoring immune function [123]. Aptamers that target PD-1 can be employed in conjunction with other immunotherapies to improve the immune response to bacterial pathogens [124]. This approach entails the prevention of interactions between PD-1 and PD-L1, which are recognized for their ability to suppress the immune response. Similar to their application in cancer immunotherapy, aptamers can facilitate the initiation of a more robust immune response against bacterial infections by inhibiting this checkpoint.

#### 4.3.4. Macrophage Polarization

Macrophages are essential in the immunological response against bacterial infection [125]. Aptamers have been designed to regulate macrophage polarization, which is a process that impacts the characteristics and activities of macrophages. For instance, aptamers that specifically attach to LPS residues hinder the process of LPS-induced macrophage polarization. They can be used as both a diagnostic tool and a therapeutic agent to reduce inflammation. Aptamer ZXL1 increases the levels of pro-inflammatory cytokines IL-1β and IL-12 while decreasing the synthesis of the anti-inflammatory cytokine IL-10 in macrophages treated with ManLAM [126]. In addition, *M.tb* specific aptamer ZXL1 increases the expression of inducible nitric oxide synthase and decreases the expression of PPAR-γ, which enhances the polarization of M1 macrophages and effectively eliminates *M*. *tb* [80].

### 4.4. Aptamer-Based Radiotherapy

Antibody-based radioimmunotherapy has proven effective in cancer treatment, and new studies indicate its potential for treating infections as well [127,128,129,130,131,132]. With the same principle, aptamer radiolabeling, the process of binding a radioactive isotope to an aptamer, allows for accurate and specific targeting of unhealthy cells, hence improving the effectiveness of therapy and minimizing adverse reactions. Radiotherapy employs high-energy radiation to induce DNA damage in sick cells or bacteria, resulting in cellular demise [133]. Presently, the investigation on aptamer radiotherapy is at a nascent phase, with limited implementation in malignancies or infections. Nevertheless, as technology continues to progress, aptamer radiation might be a very effective and accurate therapeutic approach against bacterial infections. Here, we give a concise summary of potential radionuclides against bacterial infections.

#### 4.4.1. Copper-64

Copper-64 (^64^Cu), a radioactive metal, shows potential in both therapeutic and diagnostic uses for bacterial infections [134,135]. Copper, a transition metal, is highly prevalent in the human body and is incorporated into mammalian cells by particular copper transporter proteins (CTRs) located on the cell membrane [136,137]. ^64^Cu possesses favorable physical characteristics that make it very suitable as a theranostic agent. It exhibits a half-life of 12.7 h and undergoes decay through electron capture (44%), positron emission (17%, 0.655 MeV), and beta particle emission (39%, 0.573 MeV). ^64^Cu possesses features that render it appropriate for a range of imaging or therapeutic applications, even at different dosage levels [138].

It is noteworthy to mention the study conducted by Li et al., where they created a ^64^Cu-labeled aptamer using guanine-rich oligonucleotides (GROs), specifically AS1411. This aptamer has a strong attraction to nucleolin, a surface protein that is excessively produced in numerous cancerous tumors [139]. The researchers found that the chelator CB-TE2A successfully attached the AS1411 aptamer to the radioactive metal. Subsequently, they performed experiments utilizing the radio-labeled chemical on H460 human non-small cell lung cancer cells and H460 tumor-bearing animals. The tracers exhibited satisfactory stability within living organisms and demonstrated a strong attraction to cells. Nevertheless, it was not employed for therapeutic applications.

#### 4.4.2. Lutetium-177

Lutetium-177 (^177^Lu) is a radioisotope utilized for both imaging and therapeutic purposes. It emits beta particles that can effectively eradicate tiny cancers and gamma rays that are suitable for PET imaging [140]. ^177^Lu-radiolabeled aptamers show great potential for theranostic applications because of their strong binding affinity. Aptamers, when combined with nanotechnology, particularly when integrated with stimulus-responsive nanoparticles, demonstrate manageable and specific therapeutic effects [141,142,143]. González-Ruíz et al. created modified gold nanoparticles that can specifically bind to VEGF RNA aptamers and the α(v)β(3) integrin NLS-RGD peptide and then labeled them with ^177^Lu using DOTA chelators [144]. This resulted in a combination of molecular radiotherapy and photothermal therapy, which showed synergistic effects. The treatment achieved significant tumor-killing effects both in vitro and in vivo. 

#### 4.4.3. Actinium-225

Actinium-225 (^225^Ac) is a radionuclide that emits alpha particles. It is known for its strong ability to kill cells and its potential to inhibit the growth of blood vessels. The limited distance (50–100 μm) enables precise exposure of specific cells, reducing harm to nearby healthy tissues [145]. ^225^Ac has a half-life of 10 days, resulting in the creation of many alpha-emitting offspring. The use of targeted radiotherapy is especially promising for treatment since it can provide radiation with a high degree of localization. Bandekar et al. conducted a study to assess the specific cytotoxicity of A10 PSMA aptamer-labeled targeted liposomes that contained ^225^Ac against cells expressing prostate-specific membrane antigen (PSMA) [146]. The results indicated that PSMA-targeted liposomes containing alpha particle emitters have the potential for selective anti-angiogenic alpha radiotherapy.

While aptamer-based radiotherapy is still in its early stages and primarily focused on cancer treatment, the principles underlying this approach hold promise for future applications in combating bacterial infections. Further research is needed to explore and validate these possibilities.

## 5. Challenges

In spite of their promising potential in therapeutic and diagnostic applications, aptamers encounter substantial hurdles that must be resolved to improve their performance and practical usability. An important obstacle is the intrinsic instability of aptamers in biological fluids, mostly caused by nuclease-mediated degradation. Unaltered RNA and DNA aptamers are very susceptible to destruction by nucleases found in human blood, with RNA being more fragile due to the presence of a 2′ hydroxyl group on the ribose [147,148]. The instability of these compounds restricts their effective duration and availability in the body, creating a considerable obstacle to their practical application in living organisms when used systemically. Several chemical modifications are used to improve aptamer stability. Locked Nucleic Acids (LNAs) and 2′-Fluoro or 2′-O-methyl modifications enhance thermal stability and nuclease resistance [147,149]. Spiegelmers, mirror-image aptamers, resist enzymatic degradation and are highly stable in biological fluids [150]. Capping the 3′ and 5′ ends protects against exonucleases, and phosphorothioate linkages increase resistance to endonucleases [18,59,151]. These strategies improve aptamer stability and prolong their circulation in the body.

Another obstacle is the quick elimination of aptamers from the body through the kidneys because of their small size (8–25 kDa). This diminishes their duration in the bloodstream and their potential to be absorbed by the body [152]. PEGylation refers to the process of attaching polyethylene glycol (PEG) to aptamers. This increases the aptamers’ molecular weight and decreases their removal by the kidneys and recognition by the immune system. As a result, the aptamers’ half-life is prolonged, and their pharmacokinetics are enhanced [153]. Another method that has been investigated to prolong the presence of small molecules in serum is conjugation with cholesterol [154]. 

Applying aptamers locally, such as through intra-articular injections, leverages the well-established benefits of targeted drug delivery. This method allows for the direct application of concentrated aptamers to the affected area, improving therapeutic efficacy while minimizing systemic exposure and off-target effects. Such an approach is particularly advantageous for treating localized infections, like those involving joint implants in knees and hips, where precision in drug delivery is critical. This concept aligns with the broader principles of intra-articular therapies used successfully in other medical contexts, such as osteoarthritis, where localized treatment has been shown to enhance drug efficacy and reduce adverse effects [155,156].

## 6. Conclusions and Perspectives

In this review, we analyzed the applications of aptamers in diagnosing and treating bacterial infections. Aptasensors offer high sensitivity and specificity, and labeled aptamers can precisely identify and monitor bacterial infections. These aptamers, whether used alone or in combination with antimicrobial agents, present promising solutions to the growing challenge of antibiotic resistance. However, as discussed in the previous section, several challenges remain, and those must be addressed to fully achieve the clinical potential of aptamers.

Aptamers are part of a broader landscape of antibacterial strategies, which includes drug therapies, antimicrobial peptides, and phage therapies. Each method has its unique advantages: drug therapies are well established but face the challenge of increasing antibiotic resistance [1]; antimicrobial peptides possess potent bactericidal properties but may be limited by stability and toxicity issues [98]; and phage therapies offer highly specific targeting but are often constrained by a narrow spectrum of action and regulatory challenges [156]. In contrast, aptamers offer a unique combination of high specificity, ease of modification, and lower production costs, making them particularly suitable for applications that require rapid, targeted responses, such as personalized medicine and point-of-care diagnostics. A detailed comparison of these methods has been conducted by Kalpana et al., highlighting the respective strengths and limitations of each approach [157].

Despite their promising potential, only a few aptamers have received FDA approval, and these are primarily for non-infectious diseases, such as macular degeneration (e.g., avacincaptad pegol and pegaptanib) [158,159]. The limited number of approved aptamer-based therapies can be attributed to several factors: the challenges in developing aptamers that are both stable and effective in vivo, the complexity of the SELEX process, and the rigorous regulatory pathways that must be navigated for approval. Additionally, the competition with well-established therapies like antibodies and small molecules may slow down the adoption of aptamers in clinical settings. However, with continued innovation and a focus on overcoming these hurdles, aptamers hold great promise for expanding their role in bacterial diagnostics and therapeutics.

Future research should focus on optimizing the SELEX process, enhancing delivery systems, and developing broad-spectrum aptamers capable of targeting conserved bacterial epitopes. Interdisciplinary collaboration and strategic investment in research and development will also be crucial in overcoming current challenges and unlocking the full potential of aptamers. If these challenges are addressed, aptamers could become central tools in combating bacterial infections, offering a promising alternative to traditional therapies and significantly improving patient outcomes.

## Figures and Tables

**Figure 1 pharmaceutics-16-01140-f001:**
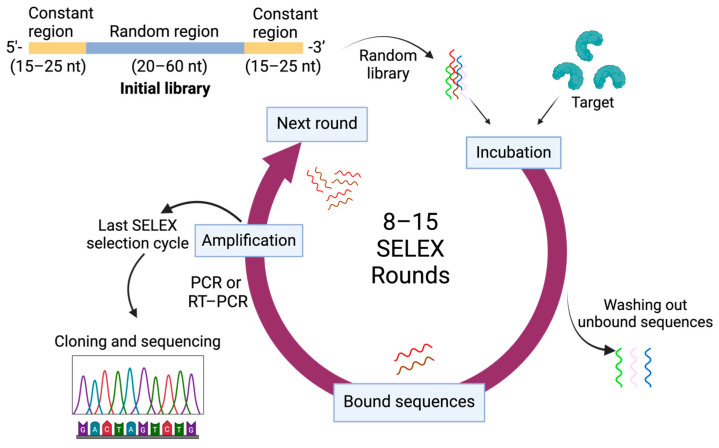
Process of Systematic Evolution of Ligands by Exponential Enrichment (SELEX).

**Figure 2 pharmaceutics-16-01140-f002:**
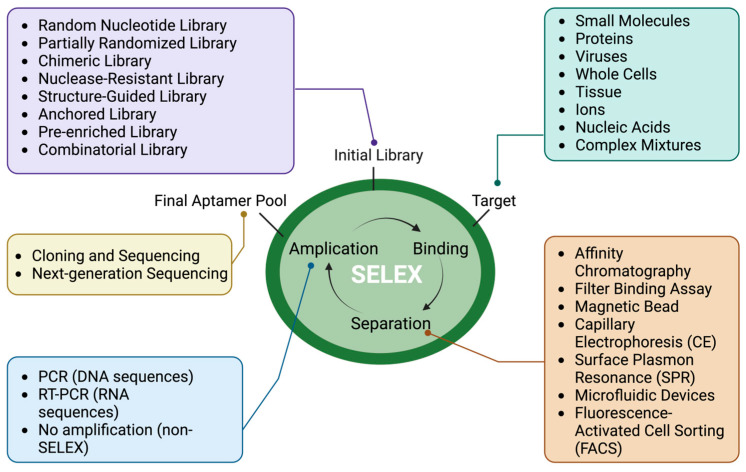
Overview of common SELEX variations.

**Figure 3 pharmaceutics-16-01140-f003:**
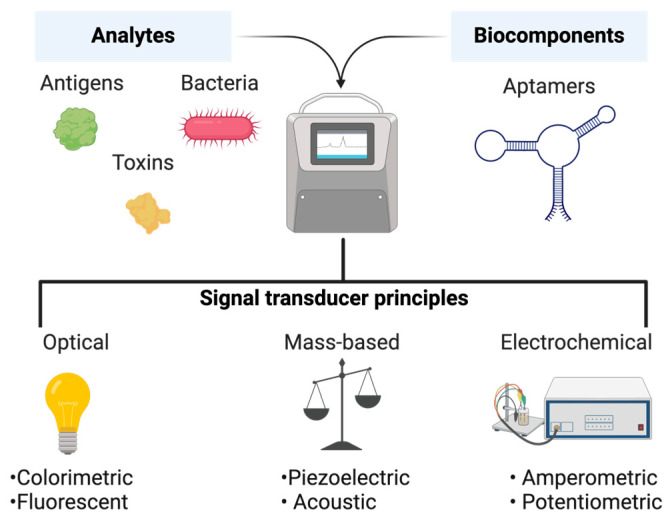
Schematic showing the different types of aptasensors.

**Figure 4 pharmaceutics-16-01140-f004:**
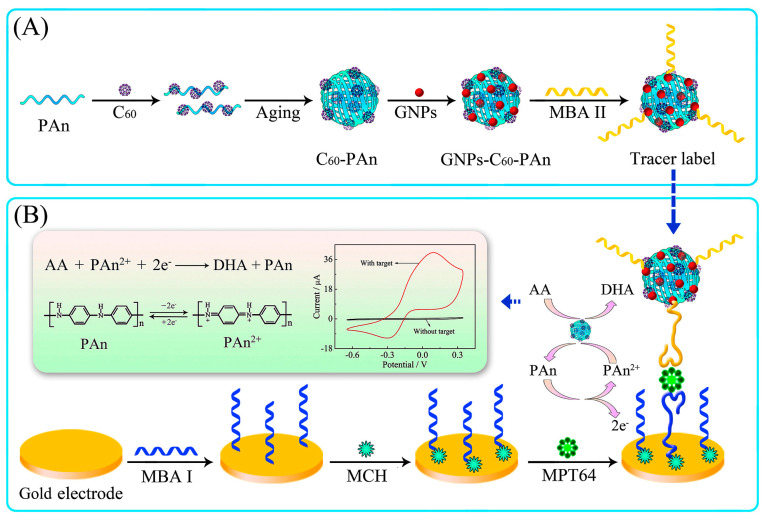
(**A**) Illustration of the preparation procedure for the tracer label. (**B**) Schematic diagram of the preparation of the electrochemical aptasensor. Reprinted with permission [46]. Copyright © 2017 Elsevier Ltd. (London, UK). The electrochemical aptasensor was used to detect the *M. tb* antigen MPT64.

**Figure 5 pharmaceutics-16-01140-f005:**
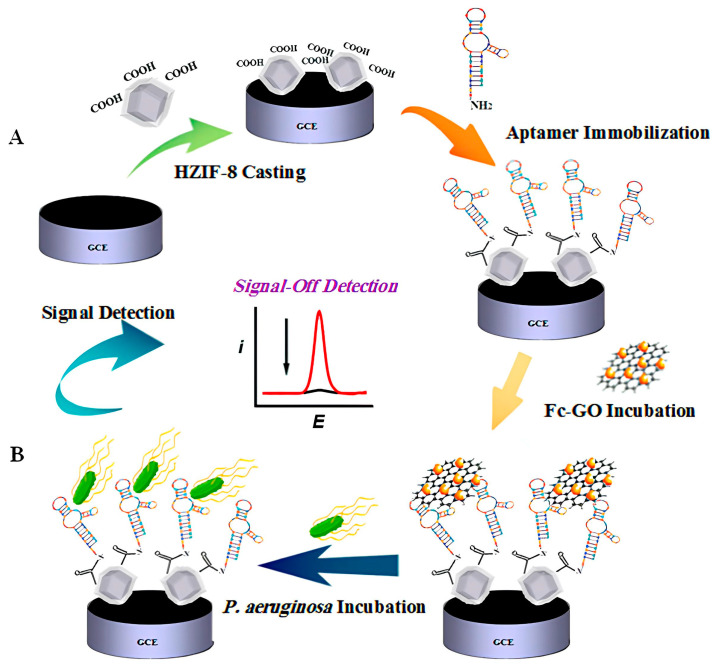
The electrochemical biosensor for detecting *P. aeruginosa*, which utilizes aptamers immobilized on a zeolitic imidazolate framework-8 (ZIFs-8) coated electrode. (**A**) Aptamers are attached to the electrode surface via EDC-NHS chemistry, followed by incubation with ferrocene–graphene oxide (Fc-GO), which serves as an electroactive indicator. (**B**) Upon exposure to *P. aeruginosa*, the aptamers bind to the bacteria, causing Fc-GO to be released from the electrode, leading to a measurable change in the electrochemical signal. Reprinted with permission [47]. Copyright © 2019 American Chemical Society.

**Figure 6 pharmaceutics-16-01140-f006:**
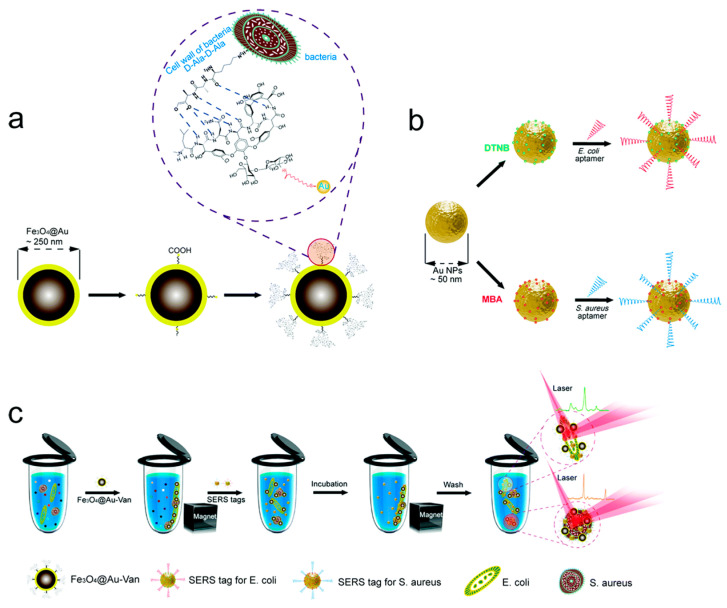
The dual-recognition platform using vancomycin and aptamers to detect both *E. coli* and *S. aureus*. (**a**) Synthesis of vancomycin-modified Fe_3_O_4_@Au MNPs, (**b**) synthesis of two different types of aptamer-conjugated SERS tags, and (**c**) operating procedure for simultaneous detection of *E. coli* and *S. aureus* via SERS platform. Reprinted with permission [51]. Copyright © Royal Society of Chemistry 2018.

**Figure 7 pharmaceutics-16-01140-f007:**
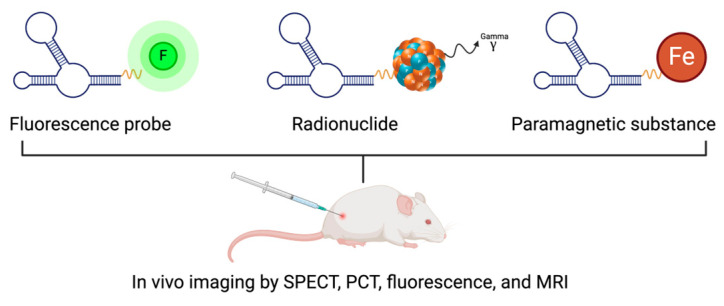
Aptamer-based in vivo molecular imaging techniques.

**Figure 8 pharmaceutics-16-01140-f008:**
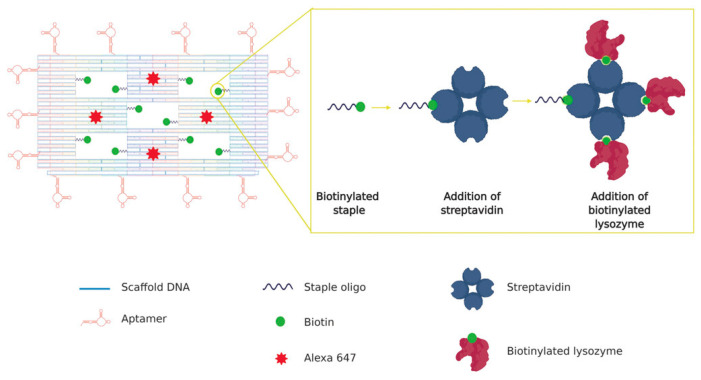
The DNA origami nanostructure with five “wells”, each carrying biotinylated staples for streptavidin attachment, followed by biotinylated lysozyme binding. The structure, equipped with 14 aptamers, targets bacteria, and four Alexa 647 molecules serve as detection markers. Reprinted with permission [108].

**Figure 9 pharmaceutics-16-01140-f009:**
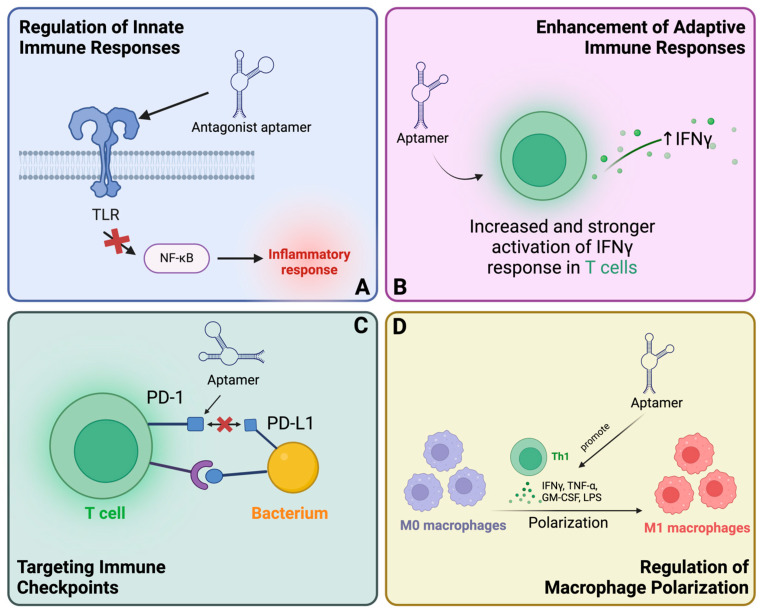
Aptamer-mediated immunomodulation in bacterial infections. (**A**) Regulation of innate immune responses; (**B**) enhancement of adaptive immune responses; (**C**) targeting immune checkpoints; (**D**) regulation of macrophage polarization.

**Table 1 pharmaceutics-16-01140-t001:** Comparison of the advantages and disadvantages of aptamers versus antibodies.

Criterion	Aptamers	Antibodies
Composition	Single-stranded DNA or RNA	Protein-based
Molecular weight	~8–25 kDa	~150–180 kDa
Specificity	Superior	High
Cost	Cost-effective	Expensive
Shelf time	Long	Short
Modifiability	Easy	Complex
Immunogenicity	Non-immunogenic	Immunogenic
Tissue penetration	Easier due to smaller size	Larger size can limit
Stability	Stable, especially with modifications	Sensitive to denaturation and degradation
Selection	In vitro SELEX	Cellular immune response in animal hosts
Production process	Chemical synthesis	In vitro biological cell culture
Target range	Broad, including small molecules and ions	Immunogenic molecules
Batch-to-batch consistency	High	Can vary

**Table 2 pharmaceutics-16-01140-t002:** Aptamer-based biosensors designed for clinical bacterial detection.

Bacterium	Aptamer Type	Target	Detection Principle	Detection Limit (LOD)	Linear Range	Detection Time	Sample Type	Reference
*Escherichia coli*, *Pseudomonas aeruginosa*	DNA	Endotoxin	Quartz crystal microbalance (QCM)	3.53 ng/mL	0–100 ng/mL	/	/	[43]
*Staphylococcus aureus*	DNA	Whole cell	Quartz crystal microbalance (QCM)	41 cfu/mL	4.1 × 10^1^–4.1 × 10^5^ cfu/mL	60 min	/	[44]
*Mycobacterium tuberculosis*	DNA	HspX	Electrochemical	10 pg	/	30 min	Cerebrospinal fluid	[45]
*Mycobacterium tuberculosis*	DNA	MPT64	Electrochemical	20 fg/mL	0.02–1000 pg/mL	/	Serum	[46]
*Pseudomonas aeruginosa*	DNA	Whole cell	Electrochemical	1 CFU/mL	1.2 × 10^1^–1.2 × 10^7^ CFU/mL	/	Urine	[47]
*Staphylococcus aureus*, *Bacillus cereus*	DNA	Whole cell	Electrochemical	100 CFU/mL	/	45 min	Milk, bovine serum	[48]
*Staphylococcus aureus*	DNA	Whole cell	Electrochemical	10⁷ CFU/mL (higher), 8 × 10² CFU/mL (lower)	/	/	/	[49]
*Staphylococcus aureus*	DNA	Whole cell	Optical (fluorescence)	1 CFU/mL	/	30 min	/	[50]
*Escherichia coli*, *Staphylococcus aureus*	DNA	Whole cell	Optical (SERS)	20 cells/mL (*S. aureus*), 50 cells/mL (*E. coli*)	/	15 min	/	[51]
*Acinetobacter baumannii*	DNA	Whole cell	Optical (fluorescence)	100 CFU/reaction	/	30 min	/	[52]

## Data Availability

Not applicable.

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
