# Peer review of "Novel Aptamer Strategies in Combating Bacterial Infections: From Diagnostics to Therapeutics"

_pharmaceutics, 2024, doi:10.3390/pharmaceutics16091140_

Round 1

Reviewer 1 Report

Comments and Suggestions for Authors

This review addresses the application of aptamers in diagnosis and therapy of bacteria caused infections. The advantages of aptamers are in their high sensitivity and specificity using SELEX libraries, the rapid adaptability to specific targets, their cost-effectiveness and long shelf life compared to antibodies. As biosensors, aptamers can transduce their response electrochemically, optically or by mass change. Several examples and application strategies are listed. Special advantages due to fast detection time exist in detection of Mycobacter tuberculosis and MRSA. Further advantages lie in therapeutic application of aptamers which are also summarized in this clearly written and comprehensive manuscript. The aptamer approach also has disadvantages and challenges, which are mentioned as well, such as inherent instability in biological fluids, e.g. due to nucleases and elimination of aptamers in the kidneys. The reference list is comprehensive and covers the major publications.

Reviewer 2 Report

Comments and Suggestions for Authors

It is basically a standard review article; personally, I was not surprised by its content, as almost all of the information is currently available in other scientific articles, but not in one compact volume. The authors focused specifically on bacterial aptamers and gave a comprehensive overview of their use. I appreciate that the authors do not repeat themselves anywhere and concentrate on a selected range of aptamers and do not waste space unnecessarily. This is the main reason why I am inclined to accept this manuscript for publication as submitted. The manuscript is very easy to read, it is very clear, the illustrations are representative and the  citations are also sufficiently relevant and up-to-date. 

Reviewer 3 Report

Comments and Suggestions for Authors

This article reviews the advances in the field of aptamers in combating bacterial infections, and the text is fluent and well-structured, making it a scientifically significant article. Therefore, I would recommend this article for publication in Pharmaceutics journal. However, before that, many concerns need to be addressed urgently and my comments are listed below.

1. Line69-70: 'Aptamers have numerous advantages compared to antibodies, such as heightened specificity...' Does the author have any evidence to support this view? It seems to me that aptamers are not as specific as antibodies. For example, there are many researchers who have reported many broad-spectrum aptamers that recognize a wide range of different targets, doesn't this reflect that aptamers are not as specific as antibodies?

2. line 130: Since the author has defined aptasensor, why not standardize the term throughout the text? The use of biosensor, aptasensor, and sometimes aptamer-based biosensor can be confusing to readers. There is a need to harmonize usage.

3. In Table 2, there is a lack of space between some of the numbers and units of measurement, for example, "10 pg".

4. In the section on detection methods, the authors only provide textual descriptions, and some of them are very brief. The reader does not understand specifically what the principle of the detection method is after reading it, for example line174-179, how did the researcher utilize the material ZIFs-8 for P. aeruginosa detection? In similar review articles, the authors provide some of the schematic diagrams (e.g., 10.1016/j.ijbiomac.2023.128677 and 10.1007/s11274-021-03002-9), so it is helpful to excerpt some of figures of the detection principle.

5. Again, there are no figures in chapters 4.2 and 4.3, and it is suggested that additional figures be added to illustrate the specific principles of these methods.

6. The paragraph structure is logically confusing, e.g., section 4.4, PDT essentially utilizes the coupling of aptamers and nanomaterials, which should be subordinated to section 4.2.

7. In chapter 4.5, although it is mentioned that Aptamer-based radiotherapy may be able to be used to treat cellular damage caused by bacteria, the application of this section is all about the treatment of cancers, and these cancers have nothing to do with bacteria. What does this have to do with the title 'Bacterial Infections'?

8. lines 608-614: Is this conclusion already supported by research? If yes, please provide references. If not, then the authors should be cautious about this.

9. The Perspectives section is too simplistic; are any aptamer-based drugs currently approved for clinical use in the treatment of bacterial infections? If not, what prevents its clinical use?

10. The authors should look beyond aptamer. Nowadays, there are more and more anti-bacterial methods other than aptamer, such as drug therapies, antimicrobial peptides, phage therapies, etc. The authors should provide a table comparing these mainstream methods. This could more effectively illustrate what advantages aptamer has.

Comments on the Quality of English Language

Moderate editing of English language required

Round 2

Reviewer 3 Report

Comments and Suggestions for Authors

Authors haver revised the submitted manuscript, and this paper can be considered to be published in the journal of Pharmaceutics.